# Correlation analysis of the proportion of monocytic myeloid-derived suppressor cells in colorectal cancer patients

**Kenna Shirasuna**[1]*, **Masayuki Ito**[1], **Takashi Matsuda**[1], **Tsuyoshi Enomoto**[2], **Yusuke Ohara**[2], **Masayoshi Yamamoto**[3], **Satomi Nishijima**[1], **Nobuhiro Ohkohchi**[2], **Sadao Kuromitsu**[1]

**1** Drug Discovery Research, Astellas Pharma, Inc., Ibaraki, Japan, **2** Department of Gastrointestinal and Hepato-Biliary-Pancreatic Surgery, University of Tsukuba Hospital, Faculty of Medicine, Ibaraki, Japan, **3** Ibaraki Western Medical Center, Ibaraki, Japan

* kenna.shirasuna@astellas.com

**Data Availability Statement:** All relevant data are within the manuscript and its Supporting information files.

## Abstract

Monocytic myeloid-derived suppressor cells (mMDSCs) are a class of immunosuppressive immune cells with prognostic value in many solid tumors. It is reported that the proportion of mMDSCs in the peripheral blood can be a predictive marker for response to cancer immunotherapy. In this study, we performed a correlation analysis of the proportion of mMDSCs in freshly-drawn peripheral blood, levels of plasma proteins, and demographic factors in colorectal cancer (CRC) patients, to find factors that could be used to predict mMDSC proportions. Freshly-drawn mMDSCs were measured using flow cytometry on peripheral blood mononuclear cells (PBMCs) from healthy donors (n = 24) and CRC patients (n = 78). The plasma concentrations of 29 different cytokines, chemokines, growth factors, and enzymes were measured using a multiplex assay or enzyme-linked immunosorbent assay. Correlation analysis to find mMDSC-associated factors was conducted using univariate and multivariate models. In univariate correlation analysis, there were no plasma proteins that were associated with mMDSC proportions in CRC patients. In multivariate analysis, considering all variables including age, sex, and plasma proteins, levels of inducible nitric acid synthase (iNOS) (p = 0.013) and platelet-derived growth factor (PDGF)-BB (p = 0.035) were associated with mMDSC proportion in PBMCs (mMDSC proportion [%] = 0.2929 − 0.2389 * PDGF-BB + 0.3582 * iNOS) (p < 0.005, r = 0.32). Measuring the plasma concentrations of iNOS and PDGF-BB may be useful in predicting the proportion of mMDSCs in CRC patients' peripheral blood. Further research is required to establish and validate these predictive factors.

Data registration

Patient data were registered in an anonymization system at Tsukuba Clinical Research & Development Organization (T-CReDO).

**Funding:** The authors declare that, other than income received from our primary employers, no financial support or compensation has been received for this research and that there are no personal conflicts of interest to declare. This research did not receive any specific grant from funding agencies in the public, commercial, or not for profit sectors. The funding organization (Astellas Pharma, Inc.) did not play a role in the study design, data collection and analysis, decision to publish, or preparation of the manuscript and only provided financial support in the form of author's salaries and/or research materials.

**Competing interests:** The funder provided support in the form of salaries for authors (KS, MI, TM, SN and SK), but did not have any additional role in the study design, data collection and analysis, decision to publish, or preparation of the manuscript. The specific roles of these authors are articulated in the 'author contributions' section. The authors declare no competing financial interests. This commercial affiliation does not alter our adherence to all PLOS ONE policies on sharing data and materials.

## Introduction

Recent data have shown that there were approximately 1.1 million new CRC cases and 551,269 CRC deaths worldwide in 2018 [1]. Previous studies have demonstrated that chronic inflammation is necessary for the initiation of CRC pathogenesis, and CRC-related inflammation promotes tumor development and progression through many different mechanisms, such as promoting angiogenesis and suppressing anti-tumor immune responses. A chronic inflammatory mucosal microenvironment can also trigger oncogenic mutations that serve as CRC-initiating events [2]. Further tumor progression is induced by inflammatory immune cells, which also work to turn an inflamed microenvironment into an immunosuppressive one [3]. It is reported that CRC induces inflammatory immune cell infiltrates through upregulation of "inflammatory signature" genes [2, 3]. Although the infiltration of CD4$^+$ T cells and CD8$^+$ T cells is associated with a good prognosis in CRC [4–6], immunosuppressive regulatory T cells and myeloid cells induce a poor prognosis [2, 3]; therefore, to characterize these immunosuppressive cells accurately is crucial for diagnosis and therapy of CRC.

Myeloid-derived suppressor cells (MDSCs), a subset of immune suppressive cells, are known to be a heterogeneous population of immature myeloid lineage cells [7, 8]. Human MDSCs are classified into two groups, CD15$^+$ granulocytic MDSCs (gMDSCs) and CD14$^+$ monocytic MDSCs (mMDSCs). Both groups of MDSCs have been shown to suppress immune responses through multiple mechanisms. These include production of NO through iNOS, release of ROS, depletion of arginine by arginase, secretion of immunosuppressive cytokines such as TGF-β and IL-10, and inducing apoptosis mediated by the Fas antigen-Fas ligand (FAS-FASL) pathway [9–14]. gMDSCs express high levels of arginase and ROS, whereas mMDSCs express high levels of both arginase and iNOS but express less ROS [15, 16]. In melanoma patients, it is reported that circulating mMDSCs suppress the NY-ESO-1 melanoma antigen-specific T cell response to tumor cells in vitro, correlate with clinical cancer stage, and show prognostic value for overall survival in stage IV disease [17]. Previous studies have also demonstrated that the number of circulating mMDSCs is significantly increased in patients with breast cancer and CRC, and correlates positively with clinical cancer stage, tumor burden, and poor clinical outcomes [18, 19].

Although measuring the proportion of peripheral mMDSCs is beneficial to predict clinical outcome in cancer patients, it requires a complex process of flow cytometric analysis with multiple cell surface markers. In addition, though mMDSCs are characterized as CD14$^+$HLA-DR$^{-/low}$ cells in humans, their HLA-DR expression typically shows wide variability, making identification of a specific subset of cells susceptible to inter-user and intra-day variability. For these reasons, measuring peripheral mMDSC levels would be difficult to incorporate as a basic clinical test. We performed a comprehensive correlation analysis including proportion of peripheral mMDSCs; multiple plasma protein levels; and demographic factors such as age, sex, and clinical grade of CRC; and demonstrated that peripheral mMDSC levels can be predicted by measuring iNOS and PDGF-BB. Although further research is required to establish and validate these predictive factors, these data suggest that measuring iNOS and PDGF-BB levels in CRC patients may be beneficial for the prediction of the clinical outcome of immunotherapy.

## Materials and methods

### Study subjects

Patients with colorectal cancer (n = 78) were recruited into this study from University of Tsukuba Hospital and Tsukuba Medical Center Hospital (Ibaraki, Japan) between April 2015 and November 2017. Prior to surgery, 20 mL of peripheral blood was collected. The inclusion

criterion was a hemoglobin concentration > 100 g/L. Exclusion criteria were viral infection with the human immunodeficiency virus, hepatitis B, or hepatitis C.

Patients were classified by disease stage according to the TNM classification system of malignant tumors published by the International Union Against Cancer. Patient data were registered in an anonymization system at Tsukuba Clinical Research & Development Organization (T-CReDO). Healthy donors (n = 24) were also recruited as a control group from employees of Astellas Pharma, Inc. (Tokyo, Japan) between December 2015 and October 2017. This study was approved by the institutional review board at University of Tsukuba Hospital (No. H26-157), Tsukuba Medical Hospital (No. 2015–036, 2016–044) and Astellas Pharma, Inc. (No. 140032, 150042, 000182), respectively. Our study was conducted in accordance with the Declaration of Helsinki. Written informed consent was obtained from all patients and healthy donors prior to blood drawing.

## PBMC isolation

Peripheral blood mononuclear cells (PBMCs) were isolated from freshly-drawn peripheral blood using BD Vacutainer CPT Mononuclear Cell Preparation Tubes (BD Bioscience, San Jose, CA, USA). The blood samples were centrifuged at 25°C for 15 minutes at 1500 rpm. The isolated PBMCs were washed twice with MACS buffer (Miltenyi Biotech, Gradbach, Germany) containing 10% bovine serum albumin. PBMCs were immediately used for flow cytometric analysis and in vitro functional assays without cryopreservation.

## Flow cytometry

PBMCs were incubated with human Fc blocker (Miltenyi Biotech) and stained with the following antibodies: Lin (CD3/CD16/CD19/CD20/CD56)-FITC, CD14-PerCP-Cy5.5, CD11b-APC-Cy7, (BD Bioscience), and HLA-DR-PE (Beckman Coulter, Brea, CA, USA) for 20 min at 4°C. The PBMCs were then washed twice with MACS buffer and then analyzed with a FACSVerse flow cytometer with FACSuite software (BD Bioscience). The subsequent data analysis was carried out using FlowJo software (Tree Star, Ashland OR, USA). The gating strategy for mMDSCs is shown in Fig 1.

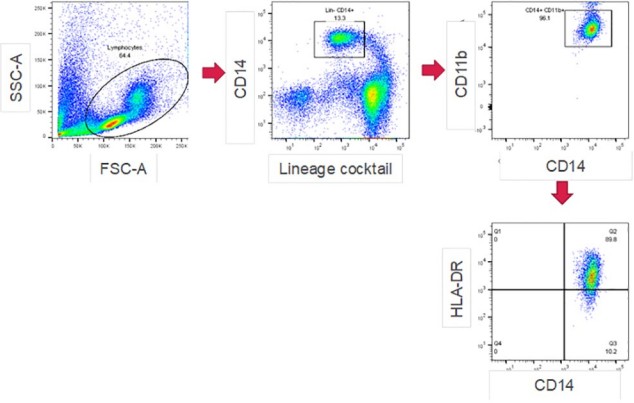

**Fig 1. Gating strategy for mMDSC detection.** Representative dot plots from flow cytometry to quantitate mMDSCs in PBMCs of healthy donors and CRC patients.

## Plasma protein measurement

Plasma was collected in the process of PBMC isolation and frozen in small aliquots at −80°C and subjected to measurement of 27 proteins (IL-1ra; IL-1β; IL-2; IL-4; IL-5; IL-6; IL-7; IL-8; IL-9; IL-10; IL-12p70; IL-13; IL-15; IL-17A; C-C motif chemokine ligand 11 [CCL11; Eotaxin]; fibroblast growth factor 2 [FGF-2]; colony stimulating factor 3 [CSF3; G-CSF]; colony stimulating factor 2 [CSF2; GM-CSF]; interferon gamma [IFN-γ]; tumor necrosis factor alpha [TNF-α]; C-X-C motif chemokine ligand 10 [CXCL10; IP-10]; C-C motif chemokine ligand 2 [CCL2; MCP-1]; C-C motif chemokine ligand 3 [CCL3; MIP-1α]; C-C motif chemokine ligand 4 [CCL4; MIP-1β]; platelet-derived growth factor-BB [PDGF-BB]; regulated on activation, normal T cell expressed and secreted [RANTES]; and vascular endothelial growth factor [VEGF]). These were measured with Bio-Plex Pro Human Cytokine Grp I Panel 27-plex (BIO RAD, Hercules, CA, USA). Arginase and iNOS were also quantified with enzyme-linked immunosorbent assays (Hycult Biotech, Uden, Netherlands and Cloud-Clone Corp, Houston, TX, USA, respectively).

## mMDSC, CD14$^{-}$ cell, and T cell isolation for in vitro mMDSC functional assay

mMDSCs were isolated by a combination of magnetic sorting and flow cytometry. The isolated PBMCs were then mixed with CD14 selection MicroBeads (Miltenyi Biotech, Monocyte Isolation Kit II) and incubated at 4°C for 15 min. The cell suspension was applied onto an LS magnetic column (Miltenyi Biotech). The column was washed with MACS buffer and unlabeled cells that passed through were collected as CD14$^{+}$ cells. HLA-DR$^{-/low}$CD14$^{+}$ cells were identified and isolated as mMDSCs using a BD FACSAria III cell sorter (BD Biosciences). The purity of the sorted populations was > 90% in all experiments. CD14$^{-}$ cells were isolated from PBMCs using CD14 selection Micro beads in parallel with CD14$^{+}$ cell isolation as above. T cells were isolated from PBMCs using human pan-T cell isolation beads (Miltenyi Biotech, Pan T Cell Isolation Kit) following the manufacturer's protocol; unlabeled cells that passed through were collected as T cells.

## In vitro mMDSC functional assay

Autologous mMDSC subsets were added at different ratios to CD14$^{-}$ cells ($5 \times 10^4$ cells/well) in 96-well flat bottom plates (Iwaki, Tokyo, Japan) in RPMI 1640 media containing 10% fetal calf serum. Cells were incubated with anti-CD2/anti-CD3/anti-CD28 antibody conjugated beads (Miltenyi Biotech) at a 1:1 CD14$^{-}$ to bead ratio. Plates were incubated at 37°C in a humidified 5% $CO_2$ incubator for 5 d. After culture, supernatants were collected and IFN-γ concentration was measured using a human IFN-γ AlphaLISA Detection Kit (Perkin Elmer, Waltham, MA, USA).

For further study, autologous mMDSC subsets ($5 \times 10^4$ cells/well) were added in a mMDSC:T cell ratio of 1:1 in 96-well flat bottom plates (Iwaki, Tokyo, Japan) in RPMI 1640 media containing 10% fetal calf serum. Cells were incubated with anti-CD2/anti-CD3/anti-CD28 antibody conjugated beads (Miltenyi Biotech) at a 1:1 T-cell to bead ratio. Plates were incubated at 37°C in a humidified 5% $CO_2$ incubator for 5 d. After culture, supernatants were collected and IFN-γ concentration was measured using a human IFN-γ AlphaLISA Detection Kit (Perkin Elmer, Waltham, MA, USA).

## Statistical analysis

When we analyzed plasma protein concentration, we excluded the measurements from the study whose value = 0 or NA for > 10 subjects, or whose 75th percentile of value was < 10

subjects. The proportion of mMDSCs in PBMCs and concentration of various plasma proteins were log-transformed for analysis. Plasma concentrations of all protein samples below the limit of quantification were assigned to 0.1 to allow log transformation. Statistical comparisons between healthy donors and CRC patients were performed using the 2-sided Welch's *t*-test for continuous variables. Statistical comparisons between mMDSC proportion and categorical variables, such as stage or the site of primary lesions in CRC patients, were conducted using a one-way analysis of variance (ANOVA). A $p < 0.05$ was chosen as the level of statistical significance for all statistical tests in this study. A prediction model against mMDSC proportion in PBMCs was developed applying a multivariate linear regression model. The variables for the regression model were selected using forward and backward stepwise feature selection method from plasma protein measurements and demographic factors (i.e., sex, and age). All statistical analyses were performed using the free statistical software R [20].

## Results

### mMDSC levels in CRC patients and healthy donors

Table 1 shows the characteristics of CRC patients and healthy donors. First, we analyzed the proportion of mMDSCs in the PBMCs. The gating strategy and the representative dot plots for mMDSCs are shown in Fig 1. The percentages of mMDSCs in the PBMCs of the 102 samples from the 78 patients with CRC and the 24 healthy donors were analyzed. The proportion of mMDSCs in the CRC patients was significantly higher than that in the healthy donors ($p < 0.001$; Fig 2).

We assessed for any correlation between mMDSC proportion with TNM tumor stage in CRC patients. CRC patients were divided into four groups based on TNM tumor stage (stage I, II, III and IV [N = 3, 21, 33 and 21, respectively]) for comparison of mMDSC proportion among the groups. Levels of mMDSCs showed no significant differences among tumor stages (Fig 3A). There was no correlation between mMDSC proportion and the site of the primary lesions in CRC patients (Fig 3B). mMDSC proportion was not gender-related (Fig 3C) or age-related (Fig 3D).

### Comparison of in vitro suppressive function of mMDSCs from CRC patients with those from healthy donors

To confirm that peripheral blood mMDSCs from CRC patients and healthy donors suppressed T-cell activation, we isolated mMDSCs, CD14⁻ cells, and pan-T cells from PBMCs and co-cultured them under stimulation with anti-CD2/anti-CD3/anti-CD28 antibody conjugated beads

**Table 1. Characteristics of study population.**

| Details of subjects | CRC patients | Healthy donors |
|---|---|---|
| Subjects (n) | 78 | 24 |
| CRC stage, (n) | | |
| I | 3 | - |
| II | 21 | - |
| III | 33 | - |
| IV | 21 | - |
| Sex, (n) | | |
| Female | 24 | 11 |
| Male | 54 | 13 |
| Age (years), median (range) | 66 (42–86) | 47.2 (40–55) |

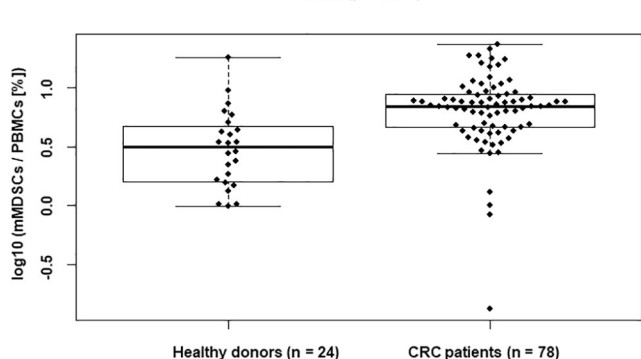

**Fig 2. Percentage of circulating mMDSCs in CRC patients and healthy donors.** mMDSC proportion in fresh PBMCs from CRC patients prior to surgery and healthy donors were analyzed. PBMCs from 78 CRC patients and 24 healthy donors were stained for mMDSC markers. mMDSC proportion (% of PBMCs) was transformed by log 10. Data were analyzed by Welch's *t*-test.

for 5 d. Then we assessed the suppressive function of mMDSCs in vitro. For mMDSC titration assay (CRC patients: n = 3, healthy donors: n = 3), when autologous mMDSCs were added in a mMDSC:CD14⁻ cell ratio of 0.25:1, 0.5:1 and 1:1, IFN-γ production of CD14⁻ cells was inhibited by mMDSCs at a ratio of 1:1 and 0.5:1, and the loss of IFN-γ suppressive activity was observed as mMDSCs were titrated down in both CRC patients and healthy donors (Fig 4A). For further assay (CRC patients: n = 9, healthy donors: n = 5), when autologous mMDSCs were co-cultured with pan-T cells at a ratio of 1:1 (mMDSCs:pan-T cells), IFN-γ production of pan-T-cells was decreased in 4 out of 5 healthy donors and 8 out of 9 CRC patients, confirming mMDSCs' suppressive function irrespective of disease state (Fig 4B; Table 2).

## Comparison of plasma protein levels between CRC patients and healthy donors

Of the target 29 plasma proteins to be measured, 15, including arginase and iNOS, were detected in CRC patients and healthy donors, and the remaining 14 proteins (IL-1b, IL-2, IL-4,

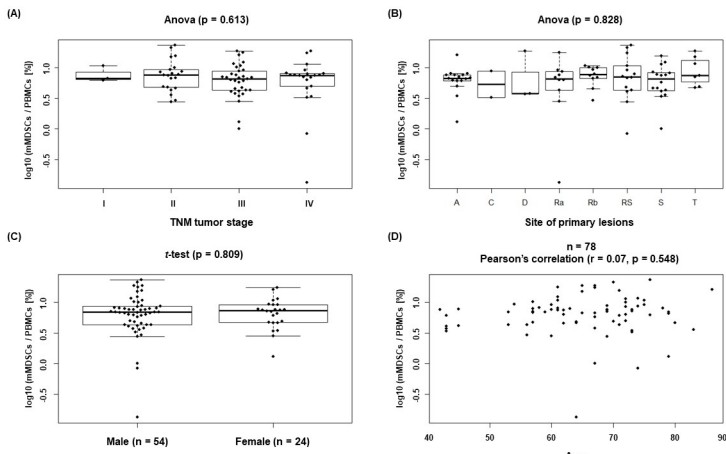

**Fig 3. Correlation analysis between mMDSC proportion and stage/demographics in CRC patients.** Correlation between mMDSC proportion and (A) TNM stage. (B) The site of the primary lesion. (C) Sex, and (D) Age. mMDSC proportion (% of PBMCs) was transformed by log 10. Data were analyzed by one-way analysis of variance (ANOVA).

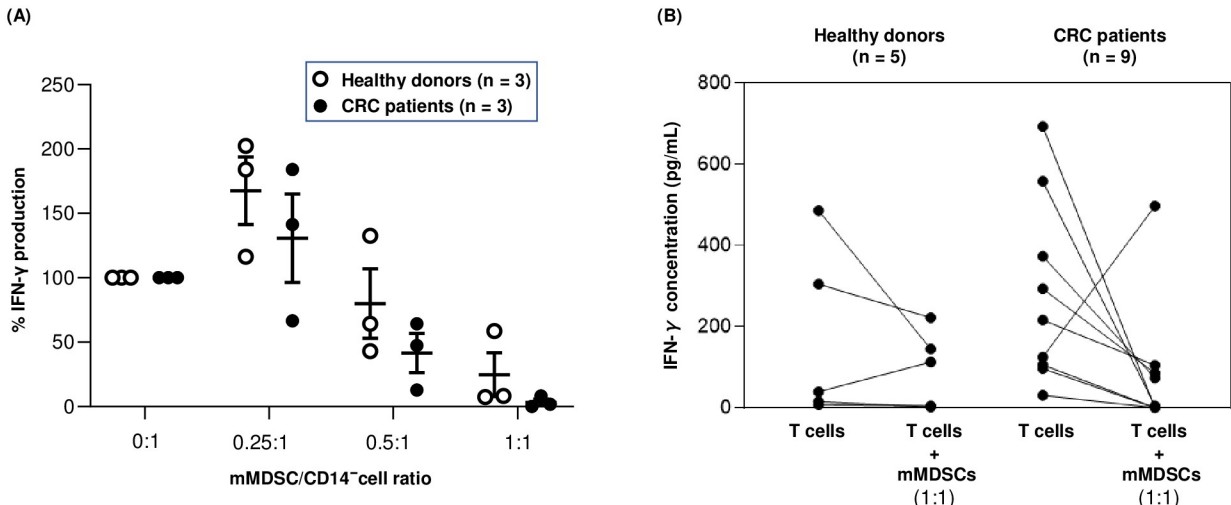

**Fig 4. In vitro suppressive activity of mMDSCs.** (A) CD14⁻ cells or (B) T cells from healthy donors and CRC patients were stimulated with anti-CD2/anti-CD3/anti-CD28 antibody conjugated beads in the absence or presence of autologous mMDSCs. Culture supernatant was collected at 5 d to measure IFN-γ concentration.

IL-5, IL-6, IL-7, IL-10, IL-12p70, IL-13, IL-15, IL-17, MIP-1a, IFN-γ and G-CSF) were excluded from analysis because their value = 0 or NA for > 10 subjects, or whose 75th percentile of value was < 10 subjects. The plasma concentrations of the 15 remaining proteins were compared between CRC patients and healthy donors. Significant differences were observed in the mean plasma levels between the CRC patients and healthy donors for 13 plasma proteins (p < 0.05, Table 3).

## Correlation analysis of mMDSC proportion in CRC patients

First, we conducted univariate correlation analysis to see if there were plasma proteins that were associated with mMDSC proportions in CRC patients, but we did not find any correlation between them (Table 4).

Next, we used multivariate methodology. A multivariate linear regression model combining forward and backward feature selection method based on AIC (Akaike Information Criterion) for mMDSC proportion was applied, considering all variables such as age, sex, and 15 plasma proteins in CRC patients. The final multivariate linear regression model included iNOS

**Table 2. Statistical analysis for suppressive function of mMDSCs.**

| | # of subjects to be tested | # of subjects showing IFN-γ reduction by mMDSCs | Mean of the differences between T cells alone and T cells + mMDSCs | p-value (two-sided paired Welch's t-test) | p-value (one-sided binominal test) |
|---|---|---|---|---|---|
| Healthy donors | 5 | 4 | −73.3 | 0.362 | 0.188 |
| CRC patients | 9 | 8 | −192.0 | 0.099 | 0.020* |
| Total | 14 | 12 | −149.6 | 0.054 | 0.006* |

* Significant at p < 0.05

**Table 3. Concentration of plasma proteins; comparisons between healthy donors and CRC patients.**

| Plasma proteins | Plasma concentration (log10: [pg/mL]) Mean (SD) | | p value (two-sided Welch's *t*-test) |
|---|---|---|---|
| | **Healthy donors** | **CRC patients** | |
| IL-1ra | 1.793 (0.281) | 1.963 (0.475) | 0.034* |
| IL-8 | 0.727 (0.739) | 1.802 (0.436) | < 0.001* |
| IL-9 | 1.275 (0.164) | 1.604 (0.363) | < 0.001* |
| Eotaxin | 1.655 (0.293) | 1.457 (0.429) | 0.013* |
| FGF basic | 1.400 (0.262) | 1.581 (0.226) | 0.004* |
| GM-CSF | 1.377 (0.811) | 1.955 (0.828) | 0.004* |
| IP-10 | 2.224 (0.174) | 2.426 (0.278) | < 0.001* |
| MCP-1 | 1.815 (0.677) | 1.851 (0.644) | 0.818 |
| PDGF-BB | 1.877 (0.333) | 2.119 (0.342) | 0.004* |
| MIP-1b | 1.515 (0.164) | 1.754 (0.168) | < 0.001* |
| RANTES | 3.077 (0.220) | 3.579 (0.405) | < 0.001* |
| TNF-α | 0.983 (0.306) | 1.201 (0.420) | 0.008* |
| VEGF | 1.487 (0.765) | 1.224 (0.681) | 0.140 |
| Arginase | 0.942 (0.400) | 1.482 (0.413) | < 0.001* |
| iNOS | 2.123 (1.488) | 2.819 (0.271) | 0.032* |

Concentration of plasma proteins (pg/mL) had been transformed by log 10

* Significant at p < 0.05

(p = 0.013) and PDGF-BB (p = 0.035) as predictive factors

$$\left[\log 10 \left(\frac{\text{mMDSCs}}{\text{PBMCs}}\right) = 0.2929 - 0.2389 * log10(PDGF - BB) + 0.3582 * \log 10(\text{iNOS})\right] (1)$$

(Pearson correlation p < 0.005 and r = 0.32) (Fig 5A).

**Table 4. Univariate analysis in CRC patients: Correlation between mMDSC proportion and plasma proteins.**

| Plasma protein | p-value | Pearson's correlation coefficient (r) |
|---|---|---|
| IL-1ra | 0.326 | −0.113 |
| IL-8 | 0.866 | −0.019 |
| IL-9 | 0.594 | −0.061 |
| Eotaxin | 0.974 | 0.004 |
| FGF basic | 0.893 | 0.015 |
| GM-CSF | 0.905 | −0.014 |
| IP-10 | 0.915 | 0.012 |
| MCP-1 | 0.826 | −0.025 |
| PDGF-BB | 0.193 | −0.149 |
| MIP-1b | 0.613 | 0.058 |
| RANTES | 0.612 | −0.058 |
| TNF-α | 0.449 | −0.087 |
| VEGF | 0.890 | 0.016 |
| Arginase | 0.794 | 0.030 |
| iNOS | 0.062 | 0.212 |

* Significant at p < 0.05

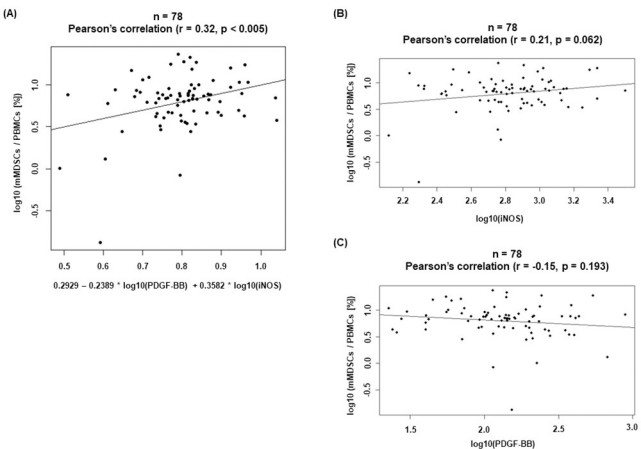

**Fig 5. Multivariate analysis for mMDSC proportion in CRC patients.** (A) A multivariate linear regression model for mMDSC proportion was constructed using several factors (plasma proteins, age, sex). Univariate analysis of mMDSC proportion and of (B) iNOS and (C) PDGF-BB concentration in CRC patients.

Of the plasma proteins selected as predictive factors in the multivariate linear regression model, plasma iNOS and PDGF-BB levels in CRC patients were significantly higher than those in healthy donors, although neither protein was associated with mMDSC proportions in CRC patients in univariate analysis (Fig 5B and 5C; Table 3).

## Discussion

In agreement with previous reports, our current work has indicated that the proportion of mMDSCs in CRC patients was significantly higher than that in healthy donors. We did not observe significant differences in the proportion of mMDSCs in different tumor stages in CRC patients. Bin Zhang and colleagues previously indicated that MDSCs in peripheral blood are associated with clinical stage and tumor metastasis in CRC patients. In their study, a significant difference was only observed between patients with advanced tumors and healthy donors while patients with stage I/II cancer had no significant increase in the proportion of circulating mMDSCs [19]. The discrepancy of MDSC proportion in CRC patients between our study and their study might be explained by the different sample processing methods and the different markers used in defining mMDSCs. They used whole blood to analyze the percentage of mMDSCs and defined mMDSCs as the Lin$^{-/low}$HLA-DR$^{-}$CD11b$^{+}$CD33$^{+}$ cells. We used PBMCs and defined mMDSCs as Lin$^{-/low}$ HLA-DR$^{-/low}$CD11b$^{+}$CD14$^{+}$ cells. We followed the staining method of mMDSCs described by Kitano and colleagues showing that peripheral mMDSC proportion correlated with overall survival in melanoma patients treated with ipili-mumab (monoclonal antibody against CTLA-4) [21]. Our study indicates that tumorigenesis significantly affects mMDSC proportion regardless of clinical stage.

Since validated specific markers for human mMDSCs are still unknown, it is important to confirm the functional activity of mMDSCs for their identification and characterization [22]. The immunosuppressive properties of mMDSCs are known [23, 24], and measuring T cell inhibition is an often-used and well-established readout system. We directly isolated mMDSCs from PBMCs in CRC patients and healthy donors by a two-step method using magnetic bead enrichment followed by flow cytometry. We first isolated CD14$^{+}$ cells by magnetic sorting and then HLA-DR$^{-/low}$ cells by flow cytometry and used these isolated cells as mMDSCs for in vitro co-culture assay with autologous CD14$^{-}$ cells or T cells isolated separately using magnetic

sorting. As a result, mMDSC-mediated suppressive activity of IFN-γ production of CD14⁻ cells increased with increasing numbers of mMDSCs in CRC patients. INF-γ production by T cells was reduced by mMDSCs in 8 out of 9 CRC patients at a 1:1 ratio. Interestingly, we found that mMDSCs showed similar suppressive activity of IFN-γ production in healthy donors. Our data suggest that the immunosuppressive activity of mMDSCs on a per-cell basis in healthy donors might be comparable to CRC patients' cells, with an increased proportion of mMDSCs in CRC patients would potentiate immune suppression.

In our study, we used fresh PBMCs for analysis since various studies emphasized the importance of using fresh blood when monitoring mMDSC proportions in the circulation [18, 25–28]. Grützner and colleagues showed that freezing PBMCs significantly decreased the yield of mMDSCs [25]. Given various studies from other groups, human mMDSCs appear to need to be studied using fresh PBMCs. However, it would be complicated to monitor mMDSC numbers from fresh PBMCs by flow cytometry with multiple-marker staining on the same day as the blood draw from patients in a clinical setting, making it desirable to establish a simple method that can predict mMDSC proportion without using fresh PBMCs. We thought that there would be value in using frozen plasma samples that can be easily measured for prediction of mMDSC proportion. We assessed whether plasma protein concentration and demographic factors such as age and sex in CRC patients could be of predictive value in terms of mMDSC proportion. The results of multivariate analysis showed that iNOS and PDGF-BB were predictive of mMDSC proportion. It is reported that mMDSCs induce increased levels of nitric oxide (NO) via iNOS leading to cell cycle arrest in T cells via depletion of the amino acid l-arginine from the tumor microenvironment [10, 29]. It is reported that angiogenic factors such as PDGF-BB have been likely involved in MDSC populations [30]. One of the potential mechanisms is direct expansion from progenitor cells by stimulation of the VEGF receptor [30, 31]. Therefore, it would be reasonable to select iNOS and PDGF-BB as predictive factors of mMDSC proportion.

Several groups have reported that mMDSCs were an important prognostic marker for cancer immunotherapy by immune checkpoint inhibitors such as ipilimumab and nivolumab (monoclonal antibodies against PD-1). In metastatic melanoma patients, mMDSC proportion was utilized to predict clinical response or resistance to ipilimumab treatment [32]. Compared with non-responders, clinical responders to ipilimumab had a significantly lower proportion of mMDSCs in the peripheral blood. This finding suggests the use of peripheral mMDSC proportion as a response marker, because a low MDSC proportion identified patients who benefitted from ipilimumab therapy [21, 32]. Other studies also reported that a lower proportion of peripheral mMDSCs at baseline can be used as a predictive marker for ipilimumab therapy for malignant melanoma [33–35]. In castration-resistant prostate cancer patients treated with prostate cancer vaccines and ipilimumab, a low mMDSC proportion in the peripheral blood was reported to be associated with clinical benefit with longer median survival [36]. Weber and colleagues indicated that a higher number of mMDSCs before treatment was associated with a poorer outcome with nivolumab in melanoma [37]. Although there is insufficient information regarding the relationship between peripheral mMDSC proportion and clinical outcome of immune checkpoint inhibitors in CRC patients, high levels of peripheral mMDSCs have been reported in this cancer [19, 38–40]. Prospective clinical trials assessing mMDSC proportions as potential biomarkers of response to immune checkpoint inhibitors in CRC patients are therefore warranted and further studies involving more CRC patients and other tumor types would be needed to validate our observations.

In conclusion, we found that iNOS and PDGF-BB are significant surrogate markers for mMDSC proportions in PBMCs in CRC patients. Our predictive model might contribute to

patient stratification in cancer immunotherapy and should guide further research on other populations with different types of malignancy.

## Supporting information

**S1 File. Information of CRC patients and healthy donors (10.6084/m9.figshare.13006922).** (XLSX)

## Acknowledgments

We thank Yuichi Iizumi and Keino Naoto for an anonymization system construction and system management of CRC patients at T-CReDO; Sunaho Kondo for technical assistance; Noboru Yamaji for critical reading for the manuscript. We thank DMC Corp. ([www.dmed.co.jp](http://www.dmed.co.jp/) <http://www.dmed.co.jp/>) for editing drafts of this manuscript.

We would like to thank the CRC patients and the healthy donors for participating in the study.

## Author Contributions

**Conceptualization:** Kenna Shirasuna, Masayuki Ito, Takashi Matsuda, Tsuyoshi Enomoto, Masayoshi Yamamoto, Satomi Nishijima, Nobuhiro Ohkohchi, Sadao Kuromitsu.

**Data curation:** Kenna Shirasuna, Masayuki Ito, Takashi Matsuda, Tsuyoshi Enomoto, Yusuke Ohara, Masayoshi Yamamoto.

**Formal analysis:** Kenna Shirasuna, Takashi Matsuda, Tsuyoshi Enomoto, Yusuke Ohara, Masayoshi Yamamoto.

**Investigation:** Kenna Shirasuna, Masayuki Ito, Tsuyoshi Enomoto, Yusuke Ohara, Masayoshi Yamamoto.

**Methodology:** Kenna Shirasuna, Masayuki Ito, Takashi Matsuda.

**Project administration:** Kenna Shirasuna.

**Resources:** Masayoshi Yamamoto, Nobuhiro Ohkohchi, Sadao Kuromitsu.

**Supervision:** Masayoshi Yamamoto, Nobuhiro Ohkohchi, Sadao Kuromitsu.

**Writing – original draft:** Kenna Shirasuna, Masayuki Ito, Takashi Matsuda.

**Writing – review & editing:** Kenna Shirasuna, Masayuki Ito, Takashi Matsuda, Satomi Nishijima, Sadao Kuromitsu.

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
