## [Decision Letter · Decision Letter 0]

13 Aug 2020

PONE-D-20-13755

Correlation analysis for the proportion of monocytic myeloid-derived suppressor cells in colorectal cancer patients

PLOS ONE

Dear Dr. Shirasuna,

Thank you for submitting your manuscript to PLOS ONE. After careful consideration, we feel that it has merit but does not fully meet PLOS ONE’s publication criteria as it currently stands. Therefore, we invite you to submit a revised version of the manuscript that addresses the points raised during the review process.

We look forward to receiving your revised manuscript.

Kind regards,

Yinyan He

Academic Editor

PLOS ONE

Journal Requirements:

2. Thank you for including your competig interests statement; "

The authors have declared that no competing interests exist"

We note that one or more of the authors are employed by a commercial company: “Astellas Pharma, Inc.”

Reviewers' comments:

Reviewer's Responses to Questions

**Comments to the Author**

1. Is the manuscript technically sound, and do the data support the conclusions?

Reviewer #1: No

2. Has the statistical analysis been performed appropriately and rigorously? 

Reviewer #1: No

3. Have the authors made all data underlying the findings in their manuscript fully available?

Reviewer #1: Yes

4. Is the manuscript presented in an intelligible fashion and written in standard English?

Reviewer #1: Yes

5. Review Comments to the Author

Reviewer #1: The topic is of clinical interest, since MDSCs have been shown to promote tumor growth in CRC and these cells are difficult to distinguish from normal myeloid cells. The main question is whether the authors’ assertion that iNOS and PDGF are reliable surrogate markers for m-MDSC frequency in the blood of CRC patients is valid. In the absence of any evidence of cause and effect, the results are purely correlational, and there are several potential reasons why the results may be misleading.

1. iNOS and PDGF might have nothing to do with mMDSC frequency, as there are many factors in blood that are different between healthy individuals and CRC patients. In their statistical analysis, have the authors applied the Bonferroni correction for multiple correlations to account for the sheer number of attempted correlations?

2. Can the authors demonstrate that the correlation does not hold true in CRC patients who have a low frequency of m-MDSC (comparable to healthy donors)?

Apart from these major concerns, the ex vivo T cell suppression assay should be performed as a dose titration, with varying numbers of MDSCs added to the co-culture. It is difficult to interpret the data without such a titration.

Also, flow plots for IFN-g secretion need to be shown

6. PLOS authors have the option to publish the peer review history of their article (what does this mean?). If published, this will include your full peer review and any attached files.

Reviewer #1: No

---

## [Author Response · Author response to Decision Letter 0]

28 Sep 2020

Dear Dr. He: 

Thank you for your email dated August 14, 2020 with your kind invitation to submit a revised version of our manuscript, “Correlation analysis of the proportion of monocytic myeloid-derived suppressor cells in colorectal cancer patients,” for further review. We have carefully reviewed the comments and have revised the manuscript accordingly. Our point-by-point responses to these comments start on the next page. 

We hope that the revised manuscript is suitable for publication in PLOS ONE and look forward to hearing from you in due course.

Journal Requirements:

https://journals.plos.org/plosone/s/file?id=wjVg/PLOSOne_formatting_sample_main_body.pdf and　https://journals.plos.org/plosone/s/file?id=ba62/PLOSOne_formatting_sample_title_authors_affiliations.pdf

Response: We followed the style template.

2. Thank you for including your competig interests statement; "

 The authors have declared that no competing interests exist"

We note that one or more of the authors are employed by a commercial company: “Astellas Pharma, Inc.”

Response: We included the following statement within our amended Funding Statement:

The authors declare that, other than income received from our primary employers, no financial support or compensation has been received for this research and that there are no personal conflicts of interest to declare. This research did not receive any specific grant from funding agencies in the public, commercial, or not for profit sectors. The funding organization (Astellas Pharma, Inc.) did not play a role in the study design, data collection and analysis, decision to publish, or preparation of the manuscript and only provided financial support in the form of author’s salaries and/or research materials. 

Response: We included the following statement within our amended Funding Statement:

The funder provided support in the form of salaries for authors (KS, MI, TM, SN and SK), but did not have any additional role in the study design, data collection and analysis, decision to publish, or preparation of the manuscript. The specific roles of these authors are articulated in the ‘author contributions’ section.

Response: We included the following statement within our amended Competing Interests Statement:

The authors declare no competing financial interests. This commercial affiliation does not alter our adherence to all PLOS ONE policies on sharing data and materials. 

Review Comments to the Author:

Reviewer #1: The topic is of clinical interest, since MDSCs have been shown to promote tumor growth in CRC and these cells are difficult to distinguish from normal myeloid cells. The main question is whether the authors’ assertion that iNOS and PDGF are reliable surrogate markers for m-MDSC frequency in the blood of CRC patients is valid. In the absence of any evidence of cause and effect, the results are purely correlational, and there are several potential reasons why the results may be misleading.

1. iNOS and PDGF might have nothing to do with mMDSC frequency, as there are many factors in blood that are different between healthy individuals and CRC patients. In their statistical analysis, have the authors applied the Bonferroni correction for multiple correlations to account for the sheer number of attempted correlations?

Response: We thank the reviewer for the helpful suggestion. The variables for the regression model were selected using forward and backward stepwise feature selection method from plasma protein measurements and demographic factors (i.e., sex and age). This feature selection method applies AIC (Akaike Information Criterion) statistics as criteria to select features/variables and does not use any statistical hypothesis test like Welch's t-test. That is, we did not select features/variables based on multiple statistical hypothesis testing of each measurement. Therefore, our opinion is that it is not necessary to apply a multiple comparison correction like the Bonferroni correction for this selection. We are sorry that this part was not clear in the original manuscript. We should have explained that our feature selection method applied AIC statistics. We added the description to the Results section as follows:

Revised (p.14, line 243-246):

Next, we used multivariate methodology. A multivariate linear regression model combining forward and backward feature selection method based on AIC (Akaike Information Criterion) for mMDSC proportion was applied, considering all variables such as age, sex, and 15 plasma proteins in CRC patients.

2. Can the authors demonstrate that the correlation does not hold true in CRC patients who have a low frequency of m-MDSC (comparable to healthy donors)?

Response: We thank the reviewer for the suggestion. We did not mean that there is no correlation in CRC patients with a low proportion of mMDSCs. Fig5A shows the relationship between predicted values (horizontal axis) and measured values (vertical axis) in the proportion of mMDSC in CRC patients. This figure reveals a trend: the lower the level of measured value, the lower is the predicted value. However, it would be difficult to present data only for CRC patients with a low frequency of mMDSCs in a statistically meaningful way due to the small number of samples. 

Reviewer #1:

Apart from these major concerns, the ex vivo T cell suppression assay should be performed as a dose titration, with varying numbers of MDSCs added to the co-culture. It is difficult to interpret the data without such a titration.

Response: We thank the reviewer for the helpful suggestion. We agree with you and have incorporated this suggestion throughout our paper. We have conducted in vitro suppression assays with titrated numbers of mMDSCs from both CRC patients (n = 3) and healthy donors (n = 3). When autologous mMDSCs were added in a mMDSC:CD14− cell ratio of 0.25:1, 0.5:1 and 1:1, IFN-γ production of CD14− cells was inhibited by mMDSCs at a ratio of 1:1 and 0.5:1, and the loss of IFN-γ suppressive activity was observed as mMDSCs were titrated down in CRC patients. We also found that mMDSCs showed the similar suppressive activity of IFN-γ production in healthy donors compared with those in CRC patients. The data are shown in Fig. 4A of the revised manuscript.

Revised

Fig4A

Revised (p.7, line 134-135)

mMDSC, CD14− cell, and T cell isolation for in vitro mMDSC functional assay

Revised (p.8, line 142-143)

CD14− cells were isolated from PBMCs using CD14 selection Micro beads in parallel with CD14+ cell isolation as above.

Revised (p.8, line 149-p.9, line 157)

Autologous mMDSC subsets were added at different ratios to CD14− cells (5 × 104 cells/well) in 96-well flat bottom plates (Iwaki, Tokyo, Japan) in RPMI 1640 media containing 10% fetal calf serum. Cells were incubated with anti-CD2/anti-CD3/anti-CD28 antibody conjugated beads (Miltenyi Biotech) at a 1:1 CD14− to bead ratio. Plates were incubated at 37°C in a humidified 5% CO2 incubator for 5 d. After culture, supernatants were collected and IFN-γ concentration was measured using a human IFN-γ AlphaLISA Detection Kit (Perkin Elmer, Waltham, MA, USA).

For further study, autologous mMDSC subsets (5 × 104 cells/well) were added in a mMDSC:T cell ratio of 1:1 in 96-well flat bottom plates (Iwaki, Tokyo, Japan) in RPMI 1640 media containing 10% fetal calf serum.

Revised (p.11, line 201-p.12, line 215)

Comparison of in vitro suppressive function of mMDSCs from CRC patients with those from healthy donors

To confirm that peripheral blood mMDSCs from CRC patients and healthy donors suppressed T-cell activation, we isolated mMDSCs, autologous CD14− cells, and pan-T cells from PBMCs and co-cultured under stimulation with anti-CD2/anti-CD3/anti-CD28 antibody conjugated beads for 5 d. Then we assessed the suppressive function of mMDSCs in vitro. For mMDSC titration assay (CRC patients: n = 3, healthy donors: n = 3), when autologous mMDSCs were added in a mMDSC:CD14− cell ratio of 0.25:1, 0.5:1 and 1:1, IFN-γ production of CD14− cells was inhibited by mMDSCs at a ratio of 1:1 and 0.5:1, and the loss of IFN-γ suppressive activity was observed as mMDSCs were titrated down in both CRC patients and healthy donors (Fig. 4A). For further assay (CRC patients: n = 9, healthy donors: n = 5), when autologous mMDSCs were co-cultured with pan-T cells at a ratio of 1:1 (mMDSCs:pan-T cells), IFN-γ production of pan-T-cells was decreased in 4 out of 5 healthy donors and 8 out of 9 CRC patients, confirming mMDSCs’ suppressive function irrespective of disease state (Fig. 4B and Table 2).

Figure 4. In vitro suppressive activity of mMDSCs 

Revised (p.16, line 279-285)

We first isolated CD14+ cells by magnetic sorting and then HLA-DR−/low cells by flow cytometry and used these isolated cells as mMDSCs for in vitro co-culture assay with autologous CD14− cells or T cells isolated separately using magnetic sorting. As a result, mMDSC-mediated suppressive activity of IFN-γ production of CD14− cells increased with increasing numbers of mMDSCs in CRC patients. INF-γ production by T cells was reduced by mMDSCs in 8 out of 9 CRC patients at a 1:1 ratio. Interestingly, we found that mMDSCs showed similar suppressive activity of IFN-γ production in healthy donors.

Revised (p.20, line 349-352)

Figure 4. In vitro suppressive activity of mMDSCs 

(A) CD14− cells or (B) T cells from healthy donors and CRC patients were stimulated with anti-CD2/anti-CD3/anti-CD28 antibody conjugated beads in the absence or presence of autologous mMDSCs. Culture supernatant was collected at 5 d to measure IFN-γ concentration.

Reviewer #1:

Also, flow plots for IFN-g secretion need to be shown

Response: We thank the reviewer for the helpful suggestion. We understand that monitoring IFN-γ secretion by flow cytometry is one of the methods in suppression assay. However, in our study, we only measured IFN-γ production by ELISA (AlphaLISA Detection Kit, Perkin Elmer, Waltham, MA, USA).

---

## [Decision Letter · Decision Letter 1]

25 Nov 2020

Correlation analysis of the proportion of monocytic myeloid-derived suppressor cells in colorectal cancer patients

PONE-D-20-13755R1

Dear Dr. Shirasuna,

We’re pleased to inform you that the revised version of your manuscript has been judged scientifically suitable for publication and will be formally accepted for publication once it meets all outstanding technical requirements.

Yours Sincerely,

Lucienne Chatenoud

Academic Editor

PLOS ONE

Additional Editor Comments (optional):

Reviewers' comments:

Reviewer's Responses to Questions

**Comments to the Author**

1. If the authors have adequately addressed your comments raised in a previous round of review and you feel that this manuscript is now acceptable for publication, you may indicate that here to bypass the “Comments to the Author” section, enter your conflict of interest statement in the “Confidential to Editor” section, and submit your "Accept" recommendation.

Reviewer #1: All comments have been addressed

2. Is the manuscript technically sound, and do the data support the conclusions?

Reviewer #1: Yes

3. Has the statistical analysis been performed appropriately and rigorously? 

Reviewer #1: Yes

4. Have the authors made all data underlying the findings in their manuscript fully available?

Reviewer #1: Yes

5. Is the manuscript presented in an intelligible fashion and written in standard English?

Reviewer #1: Yes

6. Review Comments to the Author

Reviewer #1: (No Response)

7. PLOS authors have the option to publish the peer review history of their article (what does this mean?). If published, this will include your full peer review and any attached files.

Reviewer #1: No

---

## [Editor Report · Acceptance letter]

27 Nov 2020

PONE-D-20-13755R1 

Correlation analysis of the proportion of monocytic myeloid-derived suppressor cells in colorectal cancer patients 

Dear Dr. Shirasuna:

I'm pleased to inform you that your manuscript has been deemed suitable for publication in PLOS ONE. Congratulations! Your manuscript is now with our production department. 

Kind regards, 

on behalf of

Professor Lucienne Chatenoud 

Academic Editor

PLOS ONE